# Influence of Torque on Platform Deformity of the Tri-Channel Implant: Two- and Three-Dimensional Analysis Using Micro-Computed Tomography

**DOI:** 10.3390/medicina59071311

**Published:** 2023-07-15

**Authors:** Renata Costa de Morais, Anselmo Agostinho Simionato, Izabela Cristina Maurício Moris, Graziela Bianchi Leoni, Adriana Cláudia Lapria Faria, Renata Cristina Silveira Rodrigues, Ricardo Faria Ribeiro

**Affiliations:** 1Department of Dental Materials and Prosthodontics, School of Dentistry of Ribeirão Preto, University of São Paulo, Ribeirão Preto 14040-904, Brazil; renata.costa.morais@alumini.usp.br (R.C.d.M.); anselmo.simionato@usp.br (A.A.S.); adriclalf@forp.usp.br (A.C.L.F.); renata@forp.usp.br (R.C.S.R.); 2Department of Dentistry, Ribeirão Preto University—UNAERP, Ribeirão Preto 14096-900, Brazil; izabelamoris@hotmail.com (I.C.M.M.); leoni.graziela@gmail.com (G.B.L.)

**Keywords:** dental implants, dental implant–abutment design, X-ray microtomography

## Abstract

*Background and Objectives*: The insertion of the dental implant in the bone is an essential step in prosthetic rehabilitation. The insertion torque has the potential to distort the prosthetic platform, which can cause future biomechanical problems with the continuous action of occlusal forces. The aim of this study is to evaluate different insertion torques in the deformation of tri-channel platform connections through two- and three-dimensional measurements with micro-CT. *Materials and Methods*: A total of 164 implants were divided into groups (platform diameter and type): 3.5, 3.75, and 4.3 mm NP (narrow platform), and 4.3 mm RP (regular platform). Each implant–platform group was then divided into four subgroups (*n* = 10) with different torques: T45 (45 Ncm), T80 (80 Ncm), T120 (120 Ncm), and T150 (150 Ncm). The implant–abutment–screw assemblies were scanned and the images obtained were analyzed. *Results*: A significant difference was observed for the linear and volume measures between the different platforms (*p* < 0.01) and the different implant insertion torques (*p* < 0.01). Qualitative analysis suggested a higher deformation resistance for the 3.75 NP compared to the 3.5 NP, and RP was more resistant compared to the NP. *Conclusions*: The 0.25 mm increment in the implant platform did not increase the resistance to the applied insertion torques; the 4.3 mm implant was significantly stronger compared to the 3.5 mm implant; and the proposed micro-CT analysis was considered valid for both the 2D and 3D analyses of micro-gaps, qualitatively and quantitatively.

## 1. Introduction

Osseointegrated implants have a high scientific consensus in dentistry, and great success rates are achieved with their use in oral rehabilitation [1,2] due to the synergistic combination of numerous biomechanical factors [3,4].

To overcome the disadvantages of external (loosening of the screw) and internal (low mechanical strength) hexagon connections, the tri-channel connection has emerged as an alternative, allowing for a better force distribution, reduced micro-movement, and an increased resistance to high insertion torques [5], all of which are advantages that are especially important for immediate loading cases [2,6].

In implantology, primary implant stability is decisive for a successful osseointegration process. In cases of immediate loading, there is a direct relationship between osseointegration and implant insertion torque [5], which is an excellent clinical parameter for the evaluation of primary stability [7,8]. Also, there is the platform switching concept [9,10] based on the change of the implant/abutment connection, as well as the action of extrinsic–intrinsic factors to the central region of the implant, thus promoting bone maintenance [11,12].

However, clinical investigations underlined how mechanical complications (such as instability of the implant–abutment assembly, the abutment screw loosening or fracture, and implant structural problems) generate detrimental forces between connecting structures and bone tissue that can lead to system fracture or loss, or biological problems due to bacterial infiltration in micro-gaps of the interface [13].

The modification of the bone tissue related to mechanical influence has also been the subject of focus in the literature over several years [1,2,14]. The continuous bone resorption presents clinical disadvantages for both dental implant and prostheses, which is why these parameters must be analyzed before planning dental rehabilitation in order to achieve clinical success [14,15].

Nowadays, digital technology represents a virtual access to human tissues and structures (like bone, teeth, gums, and face) in a single 3D model [12]. Hence, studies have focused on three-dimensional methods and digital dentistry (CAD/CAM) to consider the risk factors related to some of the dental implants characteristics because they influence the tension and the bone resorption and remodeling [11,14]. That is why it is fundamental to analyze and understand the stress involving the complex system around bone, dental implants, and prosthodontics components before planning any surgeries in rehabilitation [15].

Micro-computed tomography (micro-CT) is a digital technology in dentistry analysis that has been proposed for the detection and evaluation of deformations and micro-gaps between implants and abutments, as it allows for the acquisition of three-dimensional images with sample preservation [16,17]. Moreover, Scarano et al. [16] demonstrated the use of micro-CT to analyze margin discrepancies and the interface between dental implant and prosthetic components.

Thus, the objective of the present study was to evaluate different insertion torques in the deformation of tri-channel narrow platform (NP) connections and regular platform (RP) connections through two- and three-dimensional (2D and 3D) measurements with micro-CT. The hypothesis tested was that micro-CT analysis is valid for 2D and 3D micro-gap evaluations. The null hypothesis was that the deformation of different implant platforms and diameters is not affected by the insertion torque applied.

## 2. Materials and Methods

For the present study, 164 13 mm long implants with tri-channel connections produced from machined grade 4 pure titanium bar according to ASTM F67 (Dérig Bioneck, Dérig Implants, Barueri, Brazil) were divided into groups according to the platform diameter and type: 3.5, 3.75, and 4.3 mm NPs, and 4.3 mm RP (Figure 1). 

For analysis of insertion torque resistance, each implant–platform group was randomly divided into four subgroups (*n* = 10): Group T45 (45 Ncm torque), Group T80 (80 Ncm torque), Group T120 (120 Ncm torque), and Group T150 (150 Ncm torque).

Insertion torques were applied with 22 RinGrip insertion wrench (TRI, Dérig Implants, Barueri, Brazil) in a torsion machine (Biopdi, São Carlos, Brazil). One wrench per group was used for the T45 and T80 groups and one wrench per implant was used for the T120 and T150 groups, to prevent driver deformation due to higher torques. Samples were coupled to stainless steel cylinders (26 × 20 mm) with lateral screws, keeping the implant platform at the upper level of the cylinder (Figure 2). After implant positioning, the torsion machine was reset to zero and the torque was applied to the level determined for each group.

After applying the torque, abutments and their respective screws were installed in the implants with a torque of 32 Ncm using a RinGrip hex wrench (Dérig Implants, Barueri, Brazil), according to the manufacturer’s recommendations. Then, the implant–abutment–screw assemblies were analyzed by micro-CT. A new implant–platform sample was used as control for each group.

### 2.1. Micro-CT Deformation Analysis

The implant–abutment–screw assemblies were scanned on the SkyScan Model 1176 microtopography scanner (Bruker micro-CT, Kontich, Belgium) operated at 90 kV, 278 mA (0.1 mm Cu filter). A 180 degree rotation scanning was performed around the vertical axis with a rotation step of 0.5 at an isotropic resolution of 8.6 μm. The images were reconstructed using the NRecon v.1.6.9.18 (Bruker micro-CT) software, providing axial cross-sections of inner structures. The images obtained before and after applying the different torques were superimposed using DataViewer v.1.5.0 software (Bruker micro-CT). Then, 2D and 3D evaluations of the tri-channel deformation were performed using the CTAn v.1.17.7.2+ (Bruker micro-CT) software.

For the 2D evaluation, 10 equidistant sections were used in sagittal and coronal directions. Micro-gaps were measured with the Measure tool of the software in six platform regions, three in the tri-channel lobes, and three in the implant–abutment edges (Figure 3).

Sixty measurements were obtained per sample (*n* = 10), totaling 640 measurements per torque group.

For the 3D evaluation, the volume of interest (VOI, mm³) was obtained from the sharpest image of the tri-channel connection, which was in the central section of the sample. From the central section, 425 sections to the cervical direction and 425 sections to the apical direction were used as limits, totaling 850 sections per sample. The percent difference of the initial and final deformations were measured and values (µm) for each group and torque were compared with those of the new implant–abutment–screw assemblies according to the formula [5]:(1)$$\%deformation=\frac{initial\;gap-final\;gap}{initial\;gap}\times 100\;$$

### 2.2. Statistical Analysis

Comparisons between insertion torques, platform types, and differences in volume were performed by two-way analysis of variance (two-way ANOVA) using the SAS software version 9.4. The fit of the models was verified by residual analysis. For the 2D measurement, comparisons between insertion torques, platform types, and their interactions were performed through orthogonal contrasts, using the linear effect model with random effects. This model considers the repeated measurements of each specimen as a random effect, and the group and force were considered fixed effects. The fit of the model was verified by residual analysis.

## 3. Results

A significant difference was observed for the linear and volume measures between the different platforms (*p* < 0.0001) and the different implant insertion torques (*p* < 0.0001). The interaction between platform and insertion torque was significant (*p* < 0.0001), indicating that the tested platforms have different behaviors depending on the different insertion torques, as can be observed in Table 1, Table 2, Table 3 and Table 4.

Images of the micro-gap formed between the tri-channel platform and the prosthetic abutment were generated. Comparisons between the control and the experimental samples (45, 80, 120, and 150 Ncm) are shown in Figure 4.

A change in micro-gap volume was found between the tri-channel platform and prosthetic abutment, especially for smaller diameter groups (3.5 NP and 3.75 NP) at higher torques (80, 120, and 150 Ncm), with few variations among the larger diameters. The 150 Ncm insertion torque caused a significant volumetric deformation in all groups. The qualitative analysis suggested a higher deformation resistance for the 3.75 NP compared to the 3.5 NP due to the 0.25 mm increase in the body of the former. In addition, the RP was more resistant compared to the NP, with the 4.3 RP group showing no evident change in the gap in relation to the control group.

## 4. Discussion

In the present study, tri-channel dental implants with different diameters and platforms were evaluated for insertion torque resistance with micro-CT analyses to verify possible deformations. Based on the results, the test hypothesis was accepted and the null hypothesis was partially rejected.

Studies have been conducted to establish the possible correlation between implant insertion torque and primary stability [8,18]. Li et al. [18] emphasize that high insertion torques (>50 Ncm) might be beneficial for securing the implant, especially in bones of lower density. However, the authors affirm that the ideal amount of force is unclear and, as observed in the present study, significant deformations can occur in narrow platforms (3.5 NP, 3.75 NP, and 4.3 NP).

The use of micro-CT for 2D and 3D investigations of platform deformations and the adaptation of components was considered because most of the previous research regarding implant failures is directed to biological aspects, resulting in the secondary importance of reports on the mechanical causes of failure [12,19,20]. Also, digital dentistry technology improves the predictability of the treatment, providing a high-resolution CT radiology examination [16,21,22].

Significant differences in deformation were found for different platforms and insertion torques, (in both 2D and 3D analyses). In addition to the associated patient discomfort, the discrepancy between the prosthetic component and the implant causing a micro-gap at the implant–prosthetic abutment interface can lead to the penetration of microorganisms and contamination of the peri-implant tissue over time [19,23]. Regardless, the contact between an implant and a new prosthetic component is subject to micro-gaps, with values between 40 and 100 µm (depending on their characteristics) [21,24], in accordance with micrographs shown in Figure 4A–D. However, to date, there are few reports that associate failures at the implant platform level (deformation) with the adaptation of the prosthetic abutment after insertion with high torques [24,25,26].

Based on the micro-CT results showing micro-gap generations after the implant insertion using different torques, we can state that mechanical complications may occur prior to the clinical use of the dental implant [27,28]. As observed, during the insertion of the implant in the crestal bone, platform deformations might occur, leading to discrepancies of the prosthetic components (especially for narrow platform implants). This may be the cause of many of the failures reported that are only associated to microbial infiltration or bone resorption [21,22].

The results found in the present study corroborate previous studies, indicating that the adaptation of the prosthetic component and the implant insertion wrench might deform depending on the applied torque [27]. Such deformations can result in biomechanical complications over time, compromising a proper functioning and stability of implant prostheses [18,24,25].

The differences found between 3.5 NP and 3.75 NP groups indicated that the 0.25 mm reinforcement of the 3.5 mm implant was not sufficient to prevent damage to the tri-channel connector. However, the volume alteration analysis shows a smaller micro-gap with the 3.75 NP using higher torques (over 100 Ncm) compared to 3.5 NP. The results suggest that the reinforcement of the 3.75 NP could be more useful if placed in the region where the installation wrench is connected (near the cervical third) in the resistant area of the implant [24] and not in the implant body [18]. According to Maeda et al. [24], the area and thickness of the external and internal walls of the implant directly influence the resistance of the implant–installation wrench combo to the applied insertion torque and, as shown in this study, additional reinforcement should be placed nearest to the connecting area (at platform level) for the best resistance possible. Likewise, with the 4.3 mm implants, significant differences were found for deformations in the NP platform compared to RP with torques above 45 Ncm.

To eliminate or minimize micro-gaps that can lead to peri-implant problems, implant manufacturers have designed more stable connections, with more accurate adaptations between prosthetic components and implant platforms, and keeping micro-gaps in the internal region of the connections [22,26,27]. This type of connection has been considered to be of superior stability [22,28]. However, according to the results of the present study, the reduced platform was more susceptible to deformations and had a lower resistance to insertion torques greater than 80 Ncm compared to the regular platform. Similar findings were found by Kwon et al. [21] and Bambini et al. [22], where the application of 60 Ncm torques and higher also caused deformations in the platform, compromising the adaptation of the prosthetic abutment.

Cervino et al. [19] demonstrated that, according to the Federal Drug Administration FDA (in 2016), digital technology using computational modeling represents a safety and effectiveness method to predict clinical situations and, in this study, the micro-CT analysis was fundamental to quantify and qualify micro-gaps, highlighting the need to clinically investigate the effect of these prosthetic mismatches [11] and the greater susceptibility of microbial infiltration. Deformations that occurred on the implant platform were not visible to the naked eye, thus impossible to be detected by the dentist. New investigations must be carried out in order to improve and modify the physical–mechanical properties of the implants used, especially those with smaller dimensions in diameter and platform, because clinical success depends on several factors, and it is suggested to develop new equipment that can standardize the methodologies of high torque application, as well as clinical evidence to evaluate the biomechanics parameters of implants subjected to high insertion torques. Also, digital dentistry analysis has demonstrated the predictability of the patient’s planning and treatment [12,14,15]. Based on these results, thermomechanical and microbiological simulation analyses are recommended to assess the clinical implications of these deformations and their impact on the mechanical and biological failures of implants, which have usually been attributed to generic causes and reported as limitations in studies [15,19,27,28].

It is important to consider that in clinical practice, deformations of the prosthetic platform are difficult to perceive. If they occur in such a way that there are no problems with inserting the abutment or once the abutment is in position, the gap becomes impossible to see. Consequently, the lack of adaptation between the abutment and the platform, along with the increased gap, will certainly interfere with the transmission of tensions and allow for a greater flow of oral fluids and bacterial content. This can result in possibly deleterious effects as described in the literature.

One of the limitations of this study is the positioning of the implant in the matrix in a favorable condition for the prosthetic platform when applying the torques. During implant installation, there may be circumstances that change this inclination and modify the force resultant, such as tooth positioning and mouth-opening amplitude, which could cause the further deformation of the prosthetic platform and the insertion wrench. Another limitation of the study is that, with the methodology applied, it is not possible to presume the response of the bone to the applied torques and whether the deformation would be different.

Future studies on the distribution of strain around the prosthetic platform deformed by insertion torque and the reliability of abutments and prostheses installed under this condition should be conducted to provide a better understanding of the potential of a micro-gap to biomechanically influence implant behavior. The micro-gap can also influence bacterial infiltration; moreover, studies associating the insertion torque to the quantity and quality of biofilm found between the implant platform and abutment prosthetic interface could be conducted.

## 5. Conclusions

Based on the results of the present study, the 0.25 mm increment in the implant platform did not increase the resistance to the applied insertion torques; the 4.3 mm implant was significantly stronger compared to the 3.5 mm implant. In addition, the proposed micro-CT analysis was considered valid for both the 2D and 3D analyses of micro-gaps, qualitatively and quantitatively, of the new implant–abutment assemblies, which showed deformations and increased micro-gaps after insertion torque application in all tested groups.

## Figures and Tables

**Figure 1 medicina-59-01311-f001:**
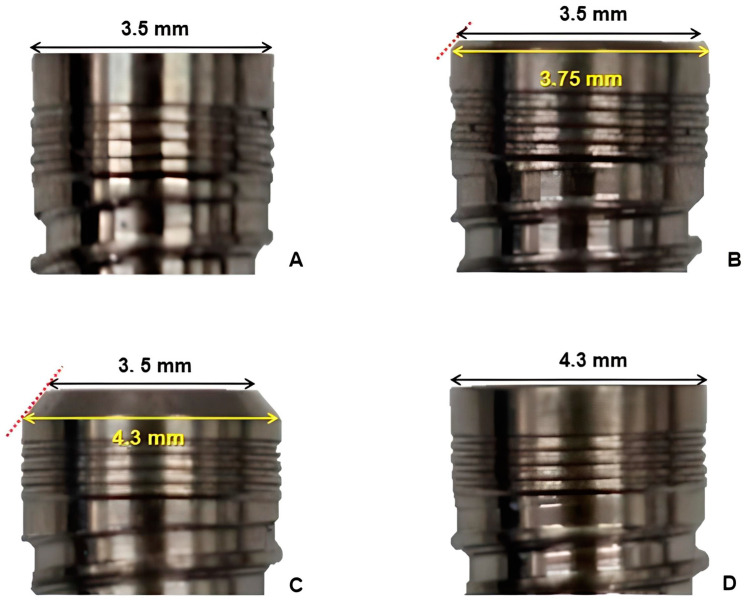
Platform and diameter sizes for implants used in the study. Black line: platform size; yellow line: diameter; red line: bevel showing the difference between platform and diameter sizes. (**A**) 3.5 narrow platform (NP) implant; (**B**) 3.75 NP implant; (**C**) 4.3 NP implant; (**D**) 4.3 regular platform (RP) implant.

**Figure 2 medicina-59-01311-f002:**
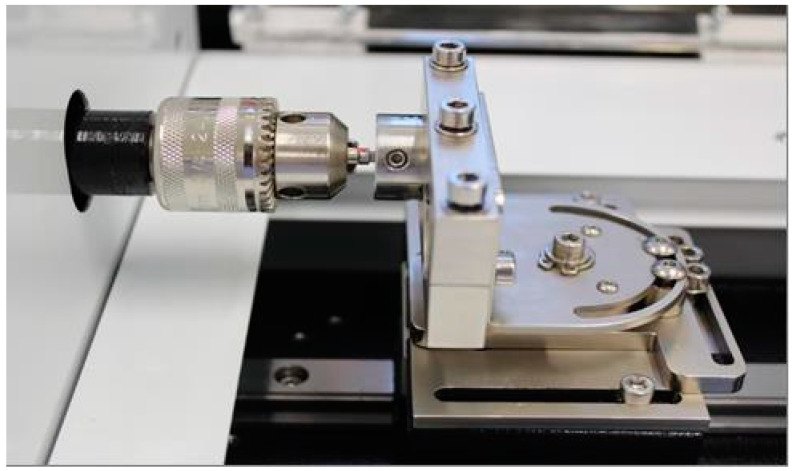
Metal cylinder implant assembly and wrench coupled to the torque machine for the application of the test insertion torques, according to the study methodology.

**Figure 3 medicina-59-01311-f003:**
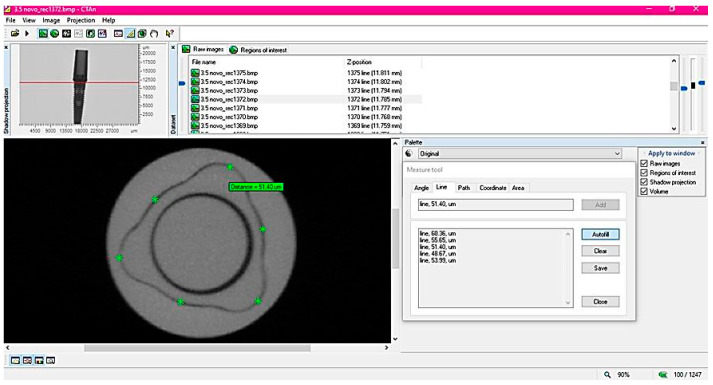
Representation of linear (two-dimensional) analysis in CT analysis. The reconstructed image (upper left corner) was fixed in the region of interest (ROI) and 10 sections were selected in that region for six measurements (asterisks) on the tri-channel platform.

**Figure 4 medicina-59-01311-f004:**
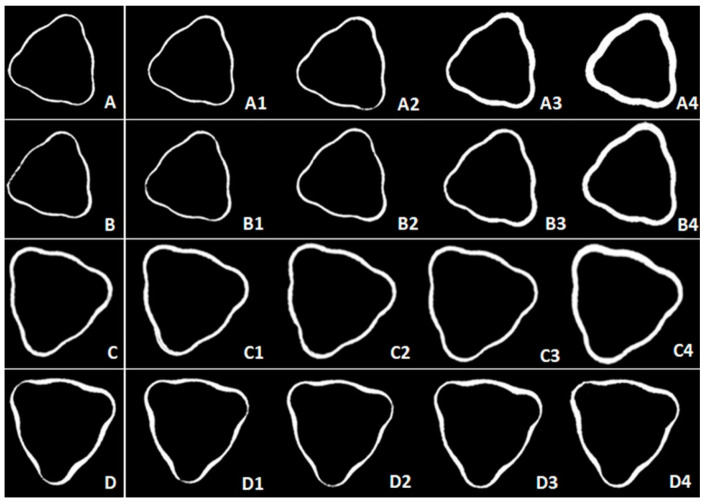
Microtopography images with the drawing of the tri-channel deformation (white area). (**A**), (**B**), (**C**), and (**D**) 3.5 NP, 3.75 NP, 4.3 NP, and 4.3 RP new implant groups (control), respectively. (**A1**), (**B1**), (**C1**), and (**D1**): images after 45 Ncm torque. (**A2**), (**B2**), (**C2**), and (**D2**): images after 80 Ncm torque. (**A3**), (**B3**), (**C3**), and (**D3**): images after 120 Ncm torque. (**A4**), (**B4**), (**C4**), and (**D4**): images after 150 Ncm torque.

**Table 1 medicina-59-01311-t001:** Comparisons of the volume (three-dimensional analysis) among insertion torques and interactions with platform types.

	Comparisons	Estimated Difference between Means	*p*-Value	Confidence Interval	
				Lower	Upper
45 Ncm	3.5 NP × 3.75 NP	4.67	<0.001	3.12	6.22
	3.5 NP × 4.3 NP	−0.10	<0.001	12.05	15.16
	3.5 NP × 4.3 RP	2.67	<0.001	12.91	16.02
	3.75 NP × 4.3 NP	13.30	<0.001	7.42	10.46
	3.75 NP × 4.3 RP	13.61	<0.001	8.28	11.31
	4.3 NP × 4.3 RP	19.52	0.270	−0.66	2.37
80 Ncm	3.5 NP × 3.75 NP	32.12	0.897	−1.62	1.42
	3.5 NP × 4.3 NP	46.58	<0.001	18.00	21.03
	3.5 NP × 4.3 RP	14.46	<0.001	21.00	24.03
	3.75 NP × 4.3 NP	22.51	<0.001	18.10	21.13
	3.75 NP × 4.3 RP	37.67	<0.001	21.10	24.13
	4.3 NP × 4.3 RP	57.49	0.000	1.48	4.51
120 Ncm	3.5 NP × 3.75 NP	8.94	0.001	1.15	4.18
	3.5 NP × 4.3 NP	19.62	<0.001	30.61	33.64
	3.5 NP × 4.3 RP	29.45	<0.001	36.16	39.19
	3.75 NP × 4.3 NP	33.29	<0.001	27.94	30.97
	3.75 NP × 4.3 RP	9.80	<0.001	33.49	36.52
	4.3 NP × 4.3 RP	22.61	<0.001	4.04	7.07
150 Ncm	3.5 NP × 3.75 NP	35.01	<0.001	11.78	14.81
	3.5 NP × 4.3 NP	44.19	<0.001	45.07	48.10
	3.5 NP × 4.3 RP	0.86	<0.001	55.98	59.01
	3.75 NP × 4.3 NP	2.30	<0.001	31.77	34.80
	3.75 NP × 4.3 RP	5.55	<0.001	42.69	45.71
	4.3 NP × 4.3 RP	10.91	<0.001	9.39	12.42

**Table 2 medicina-59-01311-t002:** Comparisons of the volume (three-dimensional analysis) among platforms and interactions with insertion torques.

	Comparisons	Estimated Difference between Means	*p*-Value	Confidence Interval	
				Lower	Upper
3.5 NP	45 Ncm × 80 Ncm	−8.53	<0.001	−10.09	−6.97
	45 Ncm × 120 Ncm	−24.72	<0.001	−26.28	−23.17
	45 Ncm × 150 Ncm	−45.77	<0.001	−47.32	−44.21
	80 Ncm × 120 Ncm	−16.19	<0.001	−17.71	−14.68
	80 Ncm × 150 Ncm	−37.24	<0.001	−38.75	−35.72
	120 Ncm × 150 Ncm	−21.04	<0.001	−22.56	−19.53
3.75 NP	45 Ncm × 80 Ncm	−13.30	<0.001	−14.81	−11.781
	45 Ncm × 120 Ncm	−26.72	<0.001	−28.24	−25.21
	45 Ncm × 150 Ncm	−37.14	<0.001	−38.65	−35.62
	80 Ncm × 120 Ncm	−13.43	<0.001	−14.94	−11.91
	80 Ncm × 150 Ncm	−23.84	<0.001	−25.35	−22.32
	120 Ncm × 150 Ncm	−10.41	<0.001	−11.93	−8.90
4.3 NP	45 Ncm × 80 Ncm	−2.62	0.001	−4.14	−1.10
	45 Ncm × 120 Ncm	−6.21	<0.001	−7.72	−4.69
	45 Ncm × 150 Ncm	−12.79	<0.001	−14.31	−11.27
	80 Ncm × 120 Ncm	−3.59	<0.001	−5.10	−2.07
	80 Ncm × 150 Ncm	−10.17	<0.001	−11.69	−8.65
	120 Ncm × 150 Ncm	−6.58	<0.001	−8.10	−5.06
4.3 RP	45 Ncm × 80 Ncm	−0.48	0.53	−2.00	1.03
	45 Ncm × 120 Ncm	−1.51	0.05	−3.03	0.00
	45 Ncm × 150 Ncm	−2.74	0.00	−4.26	−1.22
	80 Ncm × 120 Ncm	−1.03	0.18	−2.55	0.49
	80 Ncm × 150 Ncm	−2.26	0.00	−3.77	−0.74
	120 Ncm × 150 Ncm	−1.23	0.11	−2.74	0.29

**Table 3 medicina-59-01311-t003:** Comparisons of the linear measure (two-dimensional analysis) among insertion torques and interactions with platform types.

	Comparisons	Estimated Difference between Means	*p*-Value	Confidence Interval	
				Lower	Upper
45 Ncm	3.5 NP × 3.75 NP	−5.76	<0.001	−7.98	−3.55
	3.5 NP × 4.3 NP	3.55	0.002	1.33	5.77
	3.5 NP × 4.3 RP	−8.13	<0.001	−10.34	−5.91
	3.75 NP × 4.3 NP	9.31	<0.001	7.10	11.53
	3.75 NP × 4.3 RP	−2.36	0.036	−4.57	−0.15
	4.3 NP × 4.3 RP	−11.68	<0.001	−13.89	−9.46
80 Ncm	3.5 NP × 3.75 NP	−5.77	<0.001	−7.98	−3.55
	3.5 NP × 4.3 NP	4.43	<0.001	2.22	6.64
	3.5 NP × 4.3 RP	−6.859	<0.001	−9.07	−4.64
	3.75 NP × 4.3 NP	10.20	<0.001	7.98	12.41
	3.75 NP × 4.3 RP	−1.09	0.333	−3.30	1.12
	4.3 NP × 4.3 RP	−11.29	<0.001	−13.89	−9.08
120 Ncm	3.5 NP × 3.75 NP	−5.94	<0.001	−8.16	−3.73
	3.5 NP × 4.3 NP	4.46	<0.001	2.24	6.67
	3.5 NP × 4.3 RP	−1.35	0.233	−3.56	0.87
	3.75 NP × 4.3 NP	10.40	<0.001	8.19	12.61
	3.75 NP × 4.3 RP	4.59	<0.001	2.38	6.81
	4.3 NP × 4.3 RP	−5.80	<0.001	−8.02	−3.59
150 Ncm	3.5 NP × 3.75 NP	−2.87	0.011	−5.08	−0.66
	3.5 NP × 4.3 NP	8.25	<0.001	6.04	10.47
	3.5 NP × 4.3 RP	4.05	0.003	1.84	6.26
	3.75 NP × 4.3 NP	11.12	<0.001	8.91	13.34
	3.75 NP × 4.3 RP	6.92	<0.001	4.71	9.18
	4.3 NP × 4.3 RP	−4.20	0.002	−6.42	−1.99

**Table 4 medicina-59-01311-t004:** Comparisons of the linear measure (two-dimensional analysis) among insertion torques and interactions with platform types.

	Comparisons	Estimated Difference between Means	*p*-Value	Confidence Interval	
				Lower	Upper
3.5 NP	45 Ncm × 80 Ncm	−1.31	0.242	−3.49	0.88
	45 Ncm × 120 Ncm	−6.91	<0.001	−9.09	−4.72
	45 Ncm × 150 Ncm	−12.49	<0.001	−14.67	−10.30
	80 Ncm × 120 Ncm	−5.60	<0.001	−14.67	−10.30
	80 Ncm × 150 Ncm	−11.18	<0.001	−13.36	−8.99
	120 Ncm × 150 Ncm	−5.58	<0.001	−7.76	−3.39
3.75 NP	45 Ncm × 80 Ncm	−1.31	0.240	−3.49	0.87
	45 Ncm × 120 Ncm	−7.08	<0.001	−9.27	−4.90
	45 Ncm × 150 Ncm	−9.59	<0.001	−11.78	−7.41
	80 Ncm × 120 Ncm	−5.77	<0.001	−7.96	−3.59
	80 Ncm × 150 Ncm	−8.28	<0.001	−10.47	−6.10
	120 Ncm × 150 Ncm	−2.50	0.025	−4.69	−0.32
4.3 NP	45 Ncm × 80 Ncm	−0.42	0.703	−2.61	1.76
	45 Ncm × 120 Ncm	−6.00	<0.001	−8.18	−3.81
	45 Ncm × 150 Ncm	−7.78	<0.001	−9.97	−5.60
	80 Ncm × 120 Ncm	−5.57	<0.001	−7.76	−3.39
	80 Ncm × 150 Ncm	−7.36	<0.001	−9.54	−5.17
	120 Ncm × 150 Ncm	−1.78	0.110	−3.97	0.40
4.3 RP	45 Ncm × 80 Ncm	−0.04	0.972	−2.22	2.15
	45 Ncm × 120 Ncm	−0.13	0.909	−2.31	2.06
	45 Ncm × 150 Ncm	−0.31	0.782	−2.49	1.88
	80 Ncm × 120 Ncm	−0.09	0.937	−2.27	2.10
	80 Ncm × 150 Ncm	−0.27	0.809	−2.45	1.91
	120 Ncm × 150 Ncm	−0.18	0.871	−2.37	2.00

## Data Availability

The data that support the findings of this study are available from the corresponding author upon reasonable request.

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
