# Peer review of "Influence of Torque on Platform Deformity of the Tri-Channel Implant: Two- and Three-Dimensional Analysis Using Micro-Computed Tomography"

_medicina, 2023, doi:10.3390/medicina59071311_

Round 1
Reviewer 1 Report
Dear authors,
The impact of different insertion torque values in implant connections is a pertinent topic, sometimes forgotten in today's Implantology. In my opinion, the present manuscript represent an interesting input about it. Congratulations.
Even though, I outlined minor aspects of revision that might be integrated in your final re-submission:
1. In the second paragraph of Introduction, I do not agree with your perspective. It should be considered that among the "tri-channel connections" many others have emerged.
2. Authors should provide a more succint Introduction. For example, from line 60 to 78 information should be condensed.
3. Sample size calculation was not provided. The rationale for the number of specimens proposed should be justified.
4. More information regardig the titanium implant composition tested should be provided (I guess a type 5 titanium allowy was used to fabricate 164 13-mm-long implants with tri-channel connections (Dérig 91 Bioneck, Barueri, SP, Brazil) according to the platform diameter and type: 3.5, 3.75, and 4.3 mm NPs, and 4.3 mm RP.
5. "Insertion torques were applied with 22 RinGrip insertion wrench (TRI, Dérig Bi-102 oneck, Dérig) in a torsion machine"
According to this procedure, how was this machine calibrated?
6. What was the rationale to purpose a 150 Ncm test limit, since this value is infrequently achieved in the clinical setting for imediate loading.
7. I would advise authors to better discuss major limitations of your study.
8. In your opinion, which future works should follow this in vitro study to clarify your study concern about connection distortion (and a widened microgap) in implant rehabilitation sucess.
In my opinion, minor editing of English language is required.
Author Response
Responses to Reviewer 1’s comments:
The impact of different insertion torque values in implant connections is a pertinent topic, sometimes forgotten in today's Implantology. In my opinion, the present manuscript represent an interesting input about it. Congratulations.
R.: Thanks!
Even though, I outlined minor aspects of revision that might be integrated in your final re-submission:
- In the second paragraph of Introduction, I do not agree with your perspective. It should be considered that among the "tri-channel connections" many others have emerged.
R.: Our intention was to quote the opinions of other authors who placed the tri-channel connection as one of the alternatives, but we modified the text to make this clear.
- Authors should provide a more succint Introduction. For example, from line 60 to 78 information should be condensed.
R.: Modification provided.
- Sample size calculation was not provided. The rationale for the number of specimens proposed should be justified.
R.: The content of this manuscript is part of a much broader research project. The sample calculation was made for the accelerated fatigue tests and the use simulation carried out. With the objective of verifying possible alterations due to the higher insertion torques, we looked for alternatives and ended up using the microCT as described, in an unprecedented way as far as we know. Thus, as the groups were already formed, the specimens already determined for those tests were used.
- More information regardig the titanium implant composition tested should be provided (I guess a type 5 titanium allowy was used to fabricate 164 13-mm-long implants with tri-channel connections (Dérig 91 Bioneck, Barueri, SP, Brazil) according to the platform diameter and type: 3.5, 3.75, and 4.3 mm NPs, and 4.3 mm RP.
R.: Informations provided as requested.
- "Insertion torques were applied with 22 RinGrip insertion wrench (TRI, Dérig Bi-102 oneck, Dérig) in a torsion machine"
According to this procedure, how was this machine calibrated?
R.: The torsion machine was purchased for this study and was supplied by BIOPDI (Sao Carlos, SP, Brazil), a manufacturer of test equipment for various applications. The company provided us an equipment with a calibration certificate and there is a recommendation for periodic calibration made by them in maintenance service.
- What was the rationale to purpose a 150 Ncm test limit, since this value is infrequently achieved in the clinical setting for imediate loading.
R.: During a Congress of Implant Dentistry held in Sao Paulo, Brazil, in a symposium took place several manifestations of clinicians who reported applications of extremely high insertion torques to implants. Aspects related to possible deleterious effects on the adjacent bone were discussed, but there were those who categorically stated that this would not be a problem, in addition to manufacturers saying that their implants would withstand high torques. This gave rise to the idea of the evaluation we carried out for this study.
- I would advise authors to better discuss major limitations of your study.
R.: Provided as requested.
- In your opinion, which future works should follow this in vitro study to clarify your study concern about connection distortion (and a widened microgap) in implant rehabilitation sucess.
R.: Provided as requested.
In my opinion, minor editing of English language is required.
R.: Provided as requested.
Reviewer 2 Report
Comments:
The authors evaluated the effect of different insertion torques in the deformation of tri-channel platform connections through two and three-dimensional measurements with micro-CT.
From the study, the authors observed:
· A significant difference in both linear and volumetric measurements between different platforms and different insertion torques.
· Qualitative analysis suggested higher deformation resistance for the 3.75 NP compared to 25 the 3.5 NP, and RP was more resistant compared to the NP.
The authors did a great job covering the importance of this study and for their contribution to improving the literature on dental implant. The study was missing some key information:
§ Why stainless-steel cylinders were used? The elastic modulus of stainless steel is much higher than bone. Material with close resemblance to bone such as G10 would mimic the clinical scenario.
§ Figure 2 caption has a spelling error “teste”
§ How were the line measurements used to calculate deformation?
§ The text labeled 120 Ncm is missing in the table 1
§ Incorrect units of torque in Table 2, 3, and 4
§ What threshold intensity was used to visualize the volumetric deformation?
§ How was it ensured that what is observed in volumetric deformation is actual deformation and not just due to adjusting the threshold values?
§ Discussion section lacking discussion of results specially about why only certain linear deformation measurements were significant while the volumetric were not significant.
Author Response
Responses to Reviewer 2’s comments:
The authors evaluated the effect of different insertion torques in the deformation of tri-channel platform connections through two and three-dimensional measurements with micro-CT.
From the study, the authors observed:
- A significant difference in both linear and volumetric measurements between different platforms and different insertion torques.
- Qualitative analysis suggested higher deformation resistance for the 3.75 NP compared to 25 the 3.5 NP, and RP was more resistant compared to the NP.
The authors did a great job covering the importance of this study and for their contribution to improving the literature on dental implant. The study was missing some key information:
- Why stainless-steel cylinders were used? The elastic modulus of stainless steel is much higher than bone. Material with close resemblance to bone such as G10 would mimic the clinical scenario.
R.: We tried to apply the insertion torques with the implants embedded in blocks of polyurethane resin (F-16 Axson), validated as representative of the bone tissue (doi: 10.1590/s1678-77572011000100010; doi: 10.1590/s1678-77572011000300012), as done in another part of the research project. However, due to the high torques applied, the implants detached from the polyurethane and rotated. Thus, to ensure that this did not happen, we used steel cylinders.
- Figure 2 caption has a spelling error “teste”
R.: Corrected.
- How were the line measurements used to calculate deformation?
R.: As illustrated in Figure 3, after the sets were scanned and the images reconstructed in the software, a cut in the region of interest (ROI) was selected so that measurements could be taken in six platform regions, three in the tri-channel lobes and three in the implant-abutment edges. Thus, 10 implants resulted in 10 cuts and 6 measurements in each cut, totaling 640 measurements. The average of the 6 measurements represented the gap measured in each implant.
- The text labeled 120 Ncm is missing in the table 1
R.: Corrected.
- Incorrect units of torque in Table 2, 3, and 4
R.: Corrected.
- What threshold intensity was used to visualize the volumetric deformation?
R.: The threshold intensities used to visualize the volumetric deformation were:
Lower grey threshold, 103 / Upper grey threshold, 255
- How was it ensured that what is observed in volumetric deformation is actual deformation and not just due to adjusting the threshold values?
R.: The intensities applied for the threshold were standardized on the values highlighted above, with no changes at any point in the analysis. Therefore, all samples were analyzed under the same conditions, which allowed the volumetric analysis to be performed.
- Discussion section lacking discussion of results specially about why only certain linear deformation measurements were significant while the volumetric were not significant.
R.: We understand that it is still difficult to specify small variations in measurements and that the analysis method can and should be improved. Anyway, in our opinion It is important to consider that in clinical practice, deformations of the prosthetic platform are difficult to perceive. If they occur in such a way that there are no problems with inserting the abutment, and once the abutment is in position, the gap becomes impossible to see. Consequently, the lack of adaptation between the abutment and the platform, along with the increased gap, will certainly interfere with the transmission of tensions and allow for a greater flow of oral fluids and bacterial content. This can result in possible deleterious effects as described in the literature.
Reviewer 3 Report
1. What is the main question addressed by the research?
The research manuscript investigates the impact of torque and diameter on a trichannel implant through micro-CT.
2. Do you consider the topic original or relevant in the field? Does it
address a specific gap in the field?
The study addresses a need for better designing of implant components to minimize microgaps.
3. What does it add to the subject area compared with other published
material?
Contrary to most research focussing on the biological complications, the in vitro study focusses on the mechanical impact.
4. What specific improvements should the authors consider regarding the
methodology? What further controls should be considered?
English language needs moderate editing. Please check the abstract as well. Analysis looks appropriate. No further improvement is required
5. Are the conclusions consistent with the evidence and arguments presented
and do they address the main question posed?
Yes the conclusions are appropriate
6. Are the references appropriate?
Yes. May consider expanding on the discussion and adding more references.
7. Please include any additional comments on the tables and figures.
Needs formatting
moderate editing required
Author Response
Responses to Reviewer 3’s comments:
The research manuscript investigates the impact of torque and diameter on a trichannel implant through micro-CT.
- Do you consider the topic original or relevant in the field? Does it
address a specific gap in the field?
The study addresses a need for better designing of implant components to minimize microgaps.
- What does it add to the subject area compared with other published
material?
Contrary to most research focussing on the biological complications, the in vitro study focusses on the mechanical impact.
- What specific improvements should the authors consider regarding the
methodology? What further controls should be considered?
English language needs moderate editing. Please check the abstract as well. Analysis looks appropriate. No further improvement is required
- Are the conclusions consistent with the evidence and arguments presented
and do they address the main question posed?
Yes the conclusions are appropriate
- Are the references appropriate?
Yes. May consider expanding on the discussion and adding more references.
- Please include any additional comments on the tables and figures.
Needs formatting
R.: We appreciate the comments!
We tried to answer all reviewers' questions, revised the English and corrected the formatting, as suggested.
Round 2
Reviewer 2 Report
Thank you for updating the manuscript.